

# School well-being in primary school children with chronic illness. A prospective cohort study

Kathleen Schnick-Vollmer[1], Christiane Diefenbach[1], Margarete Imhof[2], Jochem König[1], Jennifer Schlecht[1], Stefan Kuhle[1] and Michael S. Urschitz[1]

[1] Institute of Medical Biostatistics, Epidemiology and Informatics, University Medical Centre of the Johannes Gutenberg-University, Mainz, Germany
[2] Institute of Psychology, Johannes Gutenberg-University, Mainz, Germany

Corresponding author
Michael S. Urschitz,
urschitz@uni-mainz.de

## ABSTRACT

**Background:** Children with chronic illness perform poorer at school, and school well-being (SWB) may mediate this association. We investigated the association between chronic illness and three domains of SWB in children in first grade.

**Methods:** Data from a German population-based prospective cohort study were used. Children with chronic illness were identified *via* their preschool health examination and follow-up parent surveys during first grade. Children were grouped as either (i) having current special health care needs (SHCN), (ii) having at least one physician diagnosis of a chronic illness but no current SHCN, or (iii) being healthy. SWB was assessed at the end of first grade and measured by the Questionnaire for the Assessment of Emotional and Social School Experiences of First and Second Grade Primary School Children. Based on SWB theory and previous frameworks, the following subscales were used: School-Related Self-Concept, Social Integration, and Joy of Learning. The sum score for each subscale was converted into area-transformed T-values (mean 50 and standard deviation 10). Associations between chronic illness groups and SWB subscales were investigated by multivariable linear regression models. Effect estimates were adjusted for potential confounding variables and standardized mean differences (SMD) were calculated.

**Results:** Of the 1,490 children included, 15% had current SHCN and 37% had a physician diagnosis of a chronic illness but no current SHCN. Compared to healthy children, children with SHCN had lower scores for the School-Related Self-Concept and the Joy of Learning subscale (SMD −0.18 for both) but not for the Social Integration subscale. In contrast, children with a chronic condition but no SHCN had lower scores only for the Social Integration subscale (SMD −0.12).

**Conclusions:** Primary school students with a chronic illness with or without SHCN have lower SWB in some domains compared to their healthy peers. SWB may be a mediator in the association between chronic illness and poor school performance.

## INTRODUCTION

Chronic illnesses during childhood represent a significant public health concern with far-reaching implications for individual health trajectories and societal burden (*Perrin, Anderson & Van Cleave, 2014*). The prevalence of chronic conditions in children has been increasing over the past decades (*Van Cleave, Gortmaker & Perrin, 2010*), with recent estimates suggesting that 16.2% of children in Germany are affected by one or more chronic health conditions (*Neuhauser, Poethko-Muller & KiGGS Study Group, 2014*). These illnesses encompass a wide range of long-term disorders including asthma, diabetes, epilepsy, and various congenital anomalies, which collectively contribute to a substantial proportion of paediatric and young adult morbidity and mortality (*Mauz, Schmitz & Poethko-Müller, 2017*).

Children with chronic conditions often face challenges that extend beyond their physical symptoms, including social development, emotional well-being, and disruptions of their education (*Pinquart & Teubert, 2012*). The recurrent nature of many chronic diseases necessitates frequent hospital visits, complex medical therapies, and sometimes invasive procedures, which can lead to increased school absenteeism and reduced academic performance (*Lum et al., 2017*). Affected children are at higher risk for anxiety, depression, and low self-esteem due to the stress associated with managing their illness and coping with potential stigmatization from peers (*Mattson & Kuo, 2019*).

In healthy children, school well-being (SWB) is a key factor in their overall well-being (*OECD, 2017*), as children spend a large part of their day in school. SWB has been linked to social skills, positive attitudes toward learning, and academic achievement; it influences a child's engagement and includes satisfaction within the school environment, which in turn affects their success in school (*Baker et al., 2003*; *Marsh & O'Mara, 2008*). Thus, SWB has been considered a key target and outcome measure in the evaluation of school health and development programs (*Lister-Sharp et al., 1999*).

Previous research has identified various factors that contribute to SWB, such as gender (*Konu & Lintonen, 2006*; *Løhre, Lydersen & Vatten, 2010*; *Løhre, Moksnes & Lillefjell, 2014*; *Markus, Rieser & Schwab, 2022*), immigrant status (*Guerra et al., 2019*), special educational needs (*Heyder, Südkamp & Steinmayr, 2020*; *Tometten, Heyder & Steinmayr, 2021*), grade (*Konu & Lintonen, 2006*; *Løhre, Lydersen & Vatten, 2010*; *Raccanello et al., 2021*), and the school environment itself (*Konu & Lintonen, 2006*; *Spratt et al., 2006*; *Weintraub & Bar-Haim Erez, 2009*). However, the contribution of chronic illness to SWB as well as the role of SWB for school performance and educational attainment in children with chronic illness has not been investigated yet.

In their analytical framework of the causal associations between health behaviour, health problems, and educational outcomes, *Suhrcke & de Paz Nieves (2011)* defined a range of possible mediators that link health exposures to educational attainment. In their model, the authors explicitly mention domains of SWB such as learning skills, discrimination in classroom, self-esteem, and quality of relationships among teachers and students. Of note, SWB was not part of the health domain but directly affected by adverse health factors; SWB in turn was considered to be a predictor for short- and long-term

educational outcomes. If these relationships can be demonstrated in practice, SWB would be a promising target for specific preventive and supportive school-based interventions in students with chronic illness.

In 2013, we commenced the ikidS Project (ikidS: ich komme in die Schule (German); I am starting school), one of the largest school health projects in Germany with a focus on chronically ill students (*Urschitz et al., 2016*). A part of the project was the ikidS Cohort Study that investigated short-and long-term effects of chronic illness in primary school children (*Hoffmann et al., 2018*). The objective of the present analysis was to examine the association between chronic illness and SWB at the end of first grade.

## MATERIALS AND METHODS

### The ikidS cohort study

Detailed information on the ikidS Project and Cohort Study has been published elsewhere (*Hoffmann et al., 2018*; *Urschitz et al., 2016*). In short, a prospective, population-based cohort study was performed between September 2014 and July 2021 within the city of Mainz and the rural district of Mainz-Bingen. The study covered a geographical area of 704 square kilometres with a population of approximately 420,000.

In 2015, there were 79 public and private primary and special needs schools in this area, and 3,683 children were registered for their first year of school. The study population consisted of all children who had their preschool health examination within the study region between September 1st, 2014, and August 31st, 2015. Participants were enrolled prior to school entry on the day of their preschool health examination. This examination is a standardised, compulsory, state-wide health examination performed by public health physicians from the regional Department of Public Health of the Mainz-Bingen District.

The participating children as well as their parents and teachers were surveyed several times: in the final year of kindergarten (T0), 6 weeks before starting school (T1), 3 months after starting school (T2), and at the end of first, third, fourth and sixth grade (T3 to T6). The preschool health examination comprised a statutory parental questionnaire (T0; including information on parental education and immigrant status), a medical history and physical examination, and the administration of screening tests as well as tests of the students' pre-academic skills. Results from the preschool health examination were provided to the investigators by the regional Department of Public Health (in personally identifiable form for study participants; in de-identified form for the total 2015 school enrolment cohort).

The cross-sectional surveys T1 to T6 of the ikidS Study collected information in the domains of general, physical and mental health, quality of life and social participation, school behaviour and performance (teachers' questionnaire), family structure and socioeconomic factors, nutritional habits and leisure time activities, and assessed the use of health care services as well as special educational support services (teachers' questionnaire).

Parental questionnaires covered different domains depending on the time points and comprised up to 20 pages and 200 items and were sent home per mail including pre-paid return envelopes. To increase the response rate, parents were sent up to three reminders to

return the questionnaires. Whenever possible, we used accepted definitions and validated items and instruments from the German Health Interview and Examination Survey for Children and Adolescents (KiGGS) (*Scheidt-Nave et al., 2007*).

Out of the 3,683 students starting first grade in the 2015/2016 school year, 2003 children (54% of the population) were recruited. Until the end of first grade, when the children were surveyed for the present study, 168 participants (5% of the population) had withdrawn from the study, moved out of the study region, or had been deferred from school enrolment, resulting in 1,835 cohort participants (50% of the population). For the present study, children were excluded if (i) they suffered from an intellectual disability, (ii) did not participate in the children's survey, or (iii) experienced comprehension difficulties during the children's survey.

The ethics committee of the State Medical Association of Rhineland-Palatinate (file # 837.544.13 (9229-F)), the regional Supervisory School Authority, and the State Representative for Data Protection in Rhineland-Palatinate approved the study. Parents or legal guardians provided written informed consent for their children.

## Physician diagnosis of a chronic illness

Information on chronic illness was obtained from the preschool health examination in the final year of kindergarten (T0) and parent surveys at T0–T3. During the preschool health examination, students are examined by school health staff and any medical, developmental, or behavioural disorders are recorded. We obtained the data for the 2015/2016 enrolment year from the Department of Public Health of the District of Mainz-Bingen.

In the parent surveys, parents were asked about the general health situation of their child, *e.g.*, "In the past 12 months, has your child experienced any of the following diseases, developmental, or behavioural disorders diagnosed by a doctor or psychologist?" The following school-relevant diagnoses were considered: anaemia, attention-deficit/hyperactivity disorder, asthma, behavioural disorders, autism, depression, diabetes, emotional problems, epilepsy, preterm birth, cardiac conditions, hay fever, hearing impairment, short stature, foetal growth restriction, atopic dermatitis, thyroid disease, vision problems, speech anomalies, cancer, snoring and sleep disorder, overweight or obesity, and underweight.

## Special health care needs

Parents were asked about their child's health care needs using the validated German version of the Children with Special Health Care Needs (SHCN) screener (*Bethell et al., 2002*; *McPherson et al., 1998*; *Schmidt & Thyen, 2008*). The screener assesses five consequences of physical or mental health conditions: (i) need for prescription medications, (ii) need for medical, social, or educational care, (iii) functional limitations, (iv) need for specialized therapies, and (v) need for treatment due to psychological issues. Each area contains a primary filter question (*e.g.*, "Does your child need or use more medical care, mental health or educational services than is usual for most children of the same age?") and one or two supplementary questions. SHCN are present when at least one

of the five filter questions and the corresponding supplementary questions were positively answered at one or more of the four time points (T0–T3).

## Study groups

The main exposure of interest–chronic illness-was categorized into three levels:

i) current SHCN regardless of the presence of a physician diagnosis of a chronic illness (children with SHCN);

ii) at least one physician diagnosis of a chronic illness but no current SHCN (children with a chronic condition but no SHCN);

iii) no physician diagnosis of a chronic illness and no current SHCN (healthy children).

## School well-being

The assessment of SWB was grounded in the Expectancy-Value Theory (*Wigfield & Eccles, 2000*) and the Self-Determination Theory (*Deci & Ryan, 2008*). Based on these theories, attitudes towards one's own abilities, enjoyment of learning and social interaction were determined as relevant domains of SWB in the context of chronic illness (*Schnick-Vollmer et al., 2020*). These domains, in turn, are assessed by the Questionnaire for the Assessment of Emotional and Social School Experiences of First and Second Grade Primary School Children (German: "Fragebogen zur Erfassung emotionaler und sozialer Schulerfahrungen von Grundschulkindern erster und zweiter Klassen" (FEESS 1–2)) (*Rauer & Schuck, 2004*). The FEESS questionnaire was specifically designed for younger students as it requires very limited reading and writing skills.

We used three subscales from the FEESS: the cognitive-motivational subscale School-Related Self-Concept, the Social Integration subscale, which includes both socio-emotional and cognitive components, and the emotions subscale Joy of Learning. The School-Related Self-Concept subscale measures both the experience of competence and the expectation of success with which a student approaches a task. The Social Integration subscale captures the personal experience of social integration. The Joy of Learning subscale represents an emotional value component that arises from the pleasure of engaging in a task or from the resulting outcome.

These subscales also fitted to the framework model for the relationship between health behaviours, health status, education, and participation by *Suhrcke & de Paz Nieves (2011)*, as well as the extensions by *Dadaczynski (2012)* and *Urschitz et al. (2016)*. In these framework models, the factors "learning abilities", "self-esteem", and "discrimination" are defined as mediators in the relationship between health aspects and educational outcomes. These factors should be approximated by the three selected subscales.

The FEESS instrument was slightly modified from the original version: First, all items were phrased positively and addressed both boys and girls equally. Second, the original layout of the FEESS documentation sheets was changed. The animal pictures used for orientation purposes in the original version were replaced by pictures of fruits to avoid emotional reactions. Third, three different questionnaire versions were created based on the order of the item blocks to prevent fatigue effects. In addition, the response options

"true" and "not true" were reworded to "yes" and "no". The adapted instrument was validated and deemed sufficiently valid for both students with and without chronic illness (*Schnick-Vollmer et al., 2020*): the reliability of the adapted FEESS subscales (Cronbach's alpha: School-Related Self-Concept: 0.84, Social Integration: 0.70, Joy of Learning: 0.83) ranged from satisfactory to good; the subscale correlations ranged from 0.58 (School-Related Self-Concept/Social Integration and Joy of Learning/Social Integration) to 0.72 (School-Related Self-Concept/Joy of Learning).

Students were asked about their SWB at the end of first grade (T3) during an in-classroom survey. The person administering the test read the statements out loud, using a neutral tone, and repeated them twice. Students responded "yes" or "no" to each item by checking the appropriate box on the questionnaire. If students required assistance, research assistants were available to support them. The study staff underwent training to keep the survey process consistent and to reduce bias.

## Confounding variables

Child, family, and institutional variables that potentially confound the association between chronic illness and SWB were identified from the adapted framework model (*Urschitz et al., 2016*) based on the original work by *Suhrcke & de Paz Nieves (2011)* and a literature review. These variables were entered into a directed acyclic graph (*Greenland, Pearl & Robins, 1999*), and a minimally sufficient adjustment set for the analysis was identified: child's gender (male/female), multiple birth (yes/no), breastfeeding duration (none/up to 6 months/6 months or more), chronic health conditions in the family (yes/no), immigrant status (child or at least one parent with foreign citizenship or born outside Germany, yes/no), socioeconomic status of the parents (ordinal) (22), attendance of the recommended paediatric check-up visits (yes/no), and school location (City of Mainz/District of Mainz-Bingen).

## Statistical analysis

All descriptive statistics were performed on complete cases only and included an analysis of representativeness of the study participants and the analysis sample compared to the study population. Distributions of continuous variables were presented as means with standard deviations (SD); categorical variables were reported as absolute and relative frequencies.

The sum score for each FEESS subscale was calculated and converted into area-transformed T-values (mean and SD of the calibration sample of 50 and 10, respectively) as recommended by the developers of the questionnaire.

To examine the relationship between chronic illness and SWB at the end of first grade, a linear mixed regression model with chronic illness as the independent variable was established for each of the three FEESS subscales. Effect estimates (beta coefficients and standardized mean differences (SMD)) were adjusted for confounding variables.

Since the experiences of students within the same class are correlated, class was considered as a random-effects variable in the models (random intercept). The models

included 168 school classes with an average of 8.9 participating children (*SD* = 3.1) per class.

Only observations with non-missing values for each FEESS subscale were included in the respective model. Missing values for the exposure and confounding variables were imputed using multivariate imputations by chained equations with 100 iterations (*van Buuren & Groothuis-Oudshoorn, 2011*).

All analyses were carried out with the statistical software package R (version 3.6.0, R Foundation for Statistical Computing, Vienna, Austria) (*R Development Core Team, 2024*).

## RESULTS

For the present study, 345 children (9% of the population) were excluded (intellectual disability (*n* = 3), non-participation in the children's survey (*n* = 309), comprehension difficulties during the children's survey (*n* = 33)), resulting in a final analysis sample size of *n* = 1490 (40% of the population).

Thirty-seven percent (*n* = 417) of the students in the analysis sample had a physician diagnosis of a chronic illness but no SHCN, while 15% (*n* = 180) had SHCN.

When comparing the analysis sample to the study population, only slight differences were observed for immigrant status and maternal education (Table 1).

Descriptive results of the FEESS subscales across study groups are shown in Table 2. Mean scores for all subscales were lower in children with chronic conditions and children with SHCN compared to healthy children.

In the multivariable regression analysis (Table 3), children with current SHCN had significantly lower scores for the School-Related Self-Concept (adjusted SMD −0.18; *p* = 0.026) and the Joy of Learning subscale compared to healthy children (adjusted SMD −0.18; *p* = 0.007).

In contrast, children with a chronic condition but no SHCN had significantly lower scores only for the Social Integration subscale (adjusted SMD −0.12; *p* = 0.038) but not for the School-Related Self-Concept (*p* = 0.69) or Joy of Learning subscales (*p* = 0.61) compared to healthy children.

## DISCUSSION

The results of the present study demonstrate that the relationship between health problems and well-being also holds true for the school setting. Students with chronic illness showed lower levels in some domains of SWB compared to their healthy peers. Our results also fit with the *a priori* assumption that SWB may be a mediator of the association between health problems and adverse educational outcomes as proposed previously (*Suhrcke & de Paz Nieves, 2011*).

The strengths of this study are the large, representative, population-based sample, the prospective design, and the use of different approaches (categorical, generic) and perspectives (school medical staff, parents) to capture a broad range of chronic illness. The concept of SWB was guided by theory, and we used a validated German instrument to assess SWB in primary school children. To our knowledge, this is the first study on the association between chronic illness and SWB despite SWB being increasingly recognized as

**Table 1 Demographic characteristics of the study population and study sample.**

| Characteristics | Study population (N = 3,683) | Study sample (n = 1,490) |
|---|---|---|
| **Child** | | |
| Gender | | |
| Male | 1,909 (51.9) | 754 (50.6) |
| Female | 1,767 (48.1) | 736 (49.4) |
| Age at preschool health examination (y), mean (SD) | 5.9 (0.4) | 5.9 (0.4) |
| Immigrant child or parent | 822 (25.5) | 316 (22.4) |
| Multiple at birth | 105 (2.9) | 45 (3.1) |
| Breastfeeding more than 6 months | 1,382 (42.3) | 630 (44.2) |
| **Parents** | | |
| Abitur (A level exams) Mother | 1,825 (60.4) | 866 (63.2) |
| Abitur (A level exams) Father | 1,768 (61.0) | 804 (61.0) |
| **School location** | | |
| District of Mainz-Bingen (rural) | 1,910 (51.9) | 780 (52.3) |
| City of Mainz (urban) | 1,773 (48.1) | 710 (47.7) |

Note:
Abbreviations: SD, standard deviation.

**Table 2 School well-being (mean and standard error) across study groups.** Results of the unadjusted regression analysis.

| School well-being subscale | n | Study groups | | |
|---|---|---|---|---|
| | | Healthy children | Children with a chronic condition | Children with special health care needs |
| School self-concept | 1,472 | 53.1 (0.40) | 52.7 (0. 49) | 50.9 (0.69) |
| Social integration | 1,474 | 51.1 (0.40) | 50.0 (0.48) | 49.6 (0.65) |
| Joy of learning | 1,473 | 52.3 (0.40) | 51.7 (0.48) | 49.7 (0.66) |

Note:
Remark: Values are reported on a T-transformed scale with mean 50 and standard deviation 10.

an important outcome in school health research. If confirmed by future studies, SWB may be evaluated as a promising target for specific school-based interventions for students with chronic illness.

We found slight differences between the two illness groups: Among children with current SHCN, both Joy of Learning and Self-Concept were negatively affected only, while among children with chronic illness but without *current* SHCN, the Social Integration domain was adversely affected. Thus, onset, severity, and timing of illness and the need for medical care may modify SWB in different ways and at different points in time. For example, a long-standing illness may have prevented social integration in kindergarten and primary school in the past but may have no impact on the present self-concept and learning, because the illness is well controlled, and the child has no current health care needs. In contrast, current SHCN may indicate an emerging or aggravating illness with immediate impact on more volatile SWB domains like self-concept and learning, while the already successful social integration is more robust and not affected. The present study,

**Table 3 Effects of chronic illness on school well-being.** Results of the adjusted multivariable regression analysis.

| School well-being subscale | n | Mean differences | | | |
| --- | --- | --- | --- | --- | --- |
| | | beta | SE | SMD | p |
| **School self-concept** | 1,486 | | | | |
| Healthy children | | Reference group | | | |
| Children with a chronic condition | | −0.25 | 0.62 | −0.02 | 0.687 |
| Children with special health care needs | | −1.79 | 0.80 | −0.18 | 0.026 |
| **Social integration** | 1,488 | | | | |
| Healthy children | | Reference group | | | |
| Children with a chronic condition | | −1.17 | 0.56 | −0.12 | 0.038 |
| Children with special health care needs | | −1.36 | 0.82 | −0.14 | 0.097 |
| **Joy of learning** | 1,487 | | | | |
| Healthy children | | Reference group | | | |
| Children with a chronic condition | | −0.29 | 0.57 | −0.03 | 0.606 |
| Children with special health care needs | | −2.21 | 0.83 | −0.18 | 0.007 |

Notes:
Remark: Effects are reported on a T-transformed scale with mean 50 and standard deviation 10.
Abbreviations: SE, standard error; SMD, standardized mean difference.

however, was small and the effect sizes were small; the results should therefore be confirmed in further studies.

Several limitations should be acknowledged: First, the health conditions considered as chronic illnesses were very heterogenous. While each of these conditions is related to various aspects of school life, their impact on SWB is likely to differ and may have led to a dilution of stronger effects for individual conditions. Future studies in larger samples should attempt to stratify the analysis by specific conditions, which was not possible in the present study.

Second, the concept of well-being is challenging to define. It differs from quality of life in that the former reflects an individual's personal evaluation of their life and experiences (*Diener, Oishi & Lucas, 2003*). The inherent subjectivity of well-being (*Dodge et al., 2012*) poses a challenge for measuring it, as researchers have to rely on self-report measures that can be influenced by personal biases and cultural differences (*Veenhoven, 2008*). Well-being is also context-specific and varies across different domains, such as work, school, family, and leisure time (*Dodge et al., 2012*). This multidimensionality required the use of a setting-specific measure that captured well-being only in school. Well-being in other settings has not been examined due to the lack of validated instruments in German. Future research should adapt available SWB instruments to other settings, develop new setting-specific instruments, and translate them into different languages.

Third, only three domains of SWB were investigated. Due to the limited attention span and reflective skills of children in first grade and time constraints imposed by school management, we had to limit the number of domains and items. Among several domain candidates provided by the FEESS instrument, we were forced to select those domains where we expected most impact by chronic illness. For example, a promising further

domain of SWB is school-related self-efficacy expectation (*Jerusalem & Satow, 1999*), a student's belief that they can handle their school tasks. Consequently, future studies should evaluate a wider range of constructs to represent SWB where possible.

Similarly, our decision to only collect information on the positive aspects of SWB might be questioned, since SWB is the result of all positive *and* negative emotions (*Diener, 2000*; *Hascher, 2004*; *Hascher, Hagenauer & Schaffner, 2011*; *Schwab et al., 2015*). We decided to not measure negative emotions and modify the corresponding items of the original FEESS instrument for two reasons: Firstly, the questions were designed to avoid deepening any existing self-critical and dysfunctional thought patterns; in particular, items like "I am a poor student" (*Rauer & Schuck, 2004*) were avoided. Secondly, children at this age have trouble grasping double negative statements (*e.g.*, "No, I am not a poor student.") (*Valtin, Wagner & Schwippert, 2005*), which may result in invalid responses for negative items.

Lastly, there is also the potential for bias in the students' responses due to individual characteristics and classroom conditions: A child may not have liked or disliked a school subject itself but the subject teacher, it may have answered socially desirable, or may have been influenced by the presence of the classroom teacher or other students. The comprehension and thus the validity of the FEESS subscales may have been influenced by factors such as the child's age, gender, or whether German was their first language. We attempted to address the influence of these individual characteristics by controlling for them in the multivariable analysis; we were, however, unable to address the potential influence of classroom conditions.

## CONCLUSIONS

Students with SHCN and, to a lesser degree, students with a chronic health condition but without SHCN have deficits in one or more aspects of SWB compared to their healthy peers. SWB may be a mediator in the association between chronic illness and poor school performance. Future research should corroborate these findings and examine the association within strata of specific chronic conditions. In addition, the influence of individual and classroom characteristics on SWB should be investigated further. Children with chronic illness and reduced SWB should be identified where possible since they are at risk of poor school performance and may benefit from specific school-based interventions. Educational interventions and social support strategies to improve SWB in children with chronic illness should be developed and evaluated.

## ACKNOWLEDGEMENTS

We wish to thank all participating parents, children, and teachers for their patience and cooperation; they made this study possible.

### Funding

This study received funding from the Federal Ministry of Education and Research of Germany (Bundesministerium für Bildung und Forschung, BMBF; Grant Agreement

Numbers 01ER1302 and 01ER1702). The funders had no role in study design, data collection and analysis, decision to publish, or preparation of the manuscript.

## Grant Disclosures
The following grant information was disclosed by the authors:
Federal Ministry of Education and Research of Germany (Bundesministerium für Bildung und Forschung, BMBF): 01ER1302, 01ER1702.

## Competing Interests
The authors declare that they have no competing interests.

## Author Contributions
- Kathleen Schnick-Vollmer conceived and designed the experiments, performed the experiments, prepared figures and/or tables, authored or reviewed drafts of the article, and approved the final draft.
- Christiane Diefenbach conceived and designed the experiments, performed the experiments, authored or reviewed drafts of the article, and approved the final draft.
- Margarete Imhof conceived and designed the experiments, authored or reviewed drafts of the article, and approved the final draft.
- Jochem König analyzed the data, authored or reviewed drafts of the article, statistical Advisor, and approved the final draft.
- Jennifer Schlecht analyzed the data, prepared figures and/or tables, and approved the final draft.
- Stefan Kuhle analyzed the data, prepared figures and/or tables, authored or reviewed drafts of the article, and approved the final draft.
- Michael S. Urschitz conceived and designed the experiments, prepared figures and/or tables, authored or reviewed drafts of the article, and approved the final draft.

## Human Ethics
The following information was supplied relating to ethical approvals (*i.e.*, approving body and any reference numbers):

The ethics committee of the State Medical Association of Rhineland-Palatinate (file # 837.544.13 (9229-F)), the regional Supervisory School Authority, and the State Representative for Data Protection in Rhineland-Palatinate approved the study.

## Data Availability
The raw data is available in the Supplemental File.

## Supplemental Information
Supplemental information for this article can be found online at http://dx.doi.org/10.7717/peerj.18280#supplemental-information.

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
