# Peer review of "School well-being in primary school children with chronic illness. A prospective cohort study"

_PeerJ, doi:10.7717/peerj.18280_

## Round 0.1 · original submission · Major Revisions

Lines 197-199: Are these Cronbach's alpha for the modified FEESS or the original instrument?

Report the intercorrelation of the subscales as well

·

Basic reporting

This is an extension of rirevious work of the group and a well written paper addressing the association of chronical illness or related special care needs on different aspects of SWB. Different aspects appear to be reduced.
The paper has several strengths: a sample of considerable size which appears to represent the study population well. Elaboration of the study instruments for adaption to the German setting.
The introduction, methods and results are concise, and the analysis is adequate. The tables are clear and the conclusions are justified.
I just have just some suggestions for the discussion:
1) Please explain why no mediator analysis regarding the role SWB on school performance is presented (a respective hint was given in the introduction).
2) If there was none the predictive properties of SWB as compared to other predictors of school performance should be addressed.

Experimental design

fine

Validity of the findings

fine

Additional comments

none

Reviewer 2 ·

Basic reporting

The overall readability of this manuscript it good, I can follow the authors logics and expressions with no problem. The reporting of the table is concise but no major issues. I have 2 comments regarding the background context:

1. The authors mentioned that "Detailed information on sampling, representativeness, and study design of the ikidS cohort has been published elsewhere" starting from line 122. Instead of leaving the readers blank information and reply on them to read other publications to get the information, I believe it would be better that authors at least provided a high-level summary of the background design. It would be at least for us the readers, to have a general idea to continue. Otherwise the flow of reading may be interrupted.

2. Starting from line 135, about the parent survey; it would be better authors can provide more information on the general setup of the survey. How many questions are there? Did the survey responses categorize into only Yes/No, or there is a scoring system for the answers to be calculated? Any information on why some parents didn't complete the survey, and on what circumstances that the parents read and complete this survey? Is it at school, or at home? Self reported outcomes usually have bias and the environment could change their response. It would be better to have more detailed information here.

Experimental design

I think the research question is clear and meaningful. This is the first time I read about any article focus on children with special needs, so I think this is a great research field. The statistical summaries are simple and direct. I have a few comments about when reporting the statistical results:

1. Starting form line 114, it would be better to have percentages after the actual count number, to have a better readability.
2. Staring from line 262, when reporting the SMD, it would be better to come with the p-value, so that the significance of the value is more obvious (or not if not significant).

Validity of the findings

Regarding the limitation of this article, the authors mentioned multiple limitations, and only 1 suggestion was observed in one of the limitations. What about the other ones? How can those limitations be reduced to minimal? What kind of other researches should be done to further evaluate or alleviate this kind of limitations? It would be better to see authors have think this through, and provide solid suggestions on how these areas can be improved.

---

## Round 0.2 · accepted · Accept

All reviewer's and editor's queries have been adequately addressed.

Reviewer 2 ·

Basic reporting

The paper is well written, the logic and readability is good. More details have been added regarding to study design background.

Experimental design

The research question is well defined.
Authors have added p-value when reporting the SMD, and the significance of the value is more direct.

Validity of the findings

Authors have elaborated the previous limitations, and how they should be address in the future. The suggestions provide valid directions for researchers to further investigate in this field.